# On-Device Training Under 256KB Memory

**Ji Lin**[1*]  **Ligeng Zhu**[1*]  **Wei-Ming Chen**[1]  **Wei-Chen Wang**[1]  **Chuang Gan**[2]  **Song Han**[1]
[1]MIT    [2]MIT-IBM Watson AI Lab
https://tinyml.mit.edu/on-device-training

## Abstract

On-device training enables the model to adapt to new data collected from the sensors by fine-tuning a pre-trained model. Users can benefit from customized AI models without having to transfer the data to the cloud, protecting the privacy. However, the training memory consumption is prohibitive for IoT devices that have tiny memory resources. We propose an algorithm-system co-design framework to make on-device training possible with only *256KB* of memory. On-device training faces two unique challenges: (1) the quantized graphs of neural networks are hard to optimize due to low bit-precision and the lack of normalization; (2) the limited hardware resource (memory and computation) does not allow full back-propagation. To cope with the optimization difficulty, we propose ***Quantization-Aware Scaling*** to calibrate the gradient scales and stabilize 8-bit quantized training. To reduce the memory footprint, we propose ***Sparse Update*** to skip the gradient computation of less important layers and sub-tensors. The algorithm innovation is implemented by a lightweight training system, ***Tiny Training Engine***, which prunes the backward computation graph to support sparse updates and offload the runtime auto-differentiation to compile time. Our framework is the *first* practical solution for on-device transfer learning of visual recognition on tiny IoT devices (*e.g.*, a microcontroller with only 256KB SRAM), using less than 1/1000 of the memory of PyTorch and TensorFlow while matching the accuracy. Our study enables IoT devices not only to perform inference but also to continuously adapt to new data for on-device lifelong learning. A video demo can be found here.

## 1 Introduction

On-device training allows us to *adapt* the pre-trained model to newly collected sensory data *after* deployment. By training and adapting *locally* on the edge, the model can learn to improve its predictions and perform lifelong learning and user customization. For example, fine-tuning a language model enables continual learning from users' typing and writing; adapting a vision model enables recognizing new objects from a mobile camera. By bringing training closer to the sensors, it also helps to protect user privacy when handling sensitive data (*e.g.*, healthcare).

However, on-device training on tiny edge devices is extremely challenging and fundamentally different from cloud training. Tiny IoT devices (*e.g.*, microcontrollers) typically have a limited SRAM size like 256KB. Such a small memory budget is hardly enough for the *inference* of deep learning models [47, 46, 7, 11, 43, 24, 44, 59], let alone the *training*, which requires extra computation for the backward and extra memory for intermediate activation [18]. On the other hand, modern deep training frameworks (*e.g.*, PyTorch [56], TensorFlow [4]) are usually designed for cloud servers and require a large memory footprint (>300MB) even when training a small model (*e.g.*, MobileNetV2-w0.35 [60]) with batch size 1 (Figure. 1).

The huge gap (>1000×) makes it impossible to run on tiny IoT devices with current frameworks and algorithms. Current deep learning training systems like PyTorch [56], TensorFlow [4], JAX [10],

---

* indicates equal contributions.

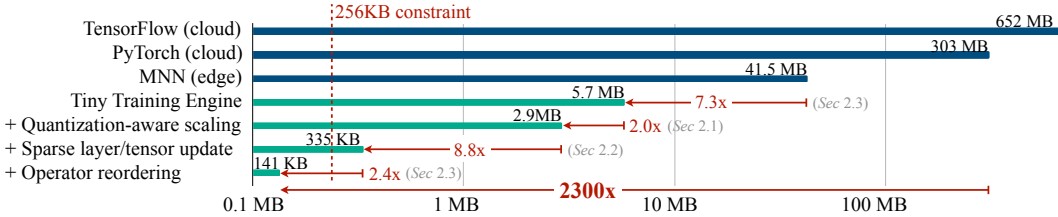

**Figure 1.** Algorithm and system co-design reduces the training memory from 303MB (PyTorch) to 141KB with the same transfer learning accuracy, leading to 2300× reduction. The numbers are measured with MobilenetV2-w0.35 [60], batch size 1 and resolution 128×128. It can be deployed to a microcontroller with 256KB SRAM.

MXNet [16], *etc.* do not consider the tight resources on edge devices. Edge deep learning inference frameworks like TVM [17], TF-Lite [3], NCNN [2], *etc.* provide a slim runtime, but lack the support for back-propagation. Though there are low-cost efficient transfer learning algorithms like training only the final classifier layer, bias-only update [12], *etc.*, the accuracy drop is significant (Figure 9), and existing training system can not realize the theoretical saving into measured saving. Furthermore, devices like microcontrollers are bare-metal and do not have an operational system and the runtime support needed by existing training frameworks. Therefore, we need to **jointly** design the *algorithm* and the *system* to enable tiny on-device training.

In this paper, we aim to bridge the gap and enable tiny on-device training with algorithm-system co-design. We investigate tiny on-device training and find two unique challenges: (1) the model is quantized on edge devices. A *real* quantized graph is difficult to optimize due to low-precision tensors and the lack of Batch Normalization layers [33]; (2) the limited hardware resource (memory and computation) of tiny hardware does not allow full back-propagation, whose memory usage can easily exceed the SRAM of microcontrollers by more than an order of magnitude. Only updating the last layer leads to poor accuracy (Figure 9). To cope with the optimization difficulty, we propose ***Quantization-Aware Scaling (QAS)*** to automatically scale the gradient of tensors with different bit-precisions, which effectively stabilizes the training and matches the accuracy of the floating-point counterpart (Section 2.1). QAS is hyper-parameter free and no tuning is required. To reduce the memory footprint of the full backward computation, we propose ***Sparse Update*** to skip the gradient computation of less important layers and sub-tensors. We developed an automated method based on contribution analysis to find the best update scheme under different memory budgets (Section 2.2). Finally, we propose a lightweight training system, ***Tiny Training Engine (TTE)*** , to implement the algorithm innovation (Section 2.3). TTE is based on code generation; it offloads the auto-differentiation to the compile-time to greatly cut down the runtime overhead. It also supports advanced graph optimization like graph pruning and reordering to support sparse updates, achieving measured memory saving and speedup.

Our framework is the **first** solution to enable tiny on-device training of convolutional neural networks under **256KB** memory budget. **(1)** Our solution enables weight update not only for the *classifier* but also for the *backbone*, which provides a *high transfer learning accuracy* (Figure 9). For tinyML application VWW [20], our on-device finetuned model matches the accuracy of cloud training+edge deployment, and surpasses the common requirement of tinyML (MLPerf Tiny [8]) by 9%. **(2)** Our system-algorithm co-design scheme effectively *reduces the memory footprint*. As shown in Figure 1, the proposed techniques greatly reduce the memory usage by more than 100× compared to the best edge training framework we can find (MNN [35]). **(3)** Our framework also greatly *accelerates training*, reducing the per-iteration time by more than 20× compared to dense update and vanilla system design (Figure 10). **(4)** We deployed our training system to a Cortex M7 microcontroller STM32F746 to demonstrate the feasibility, suggesting that tiny IoT devices can not only perform inference but also training to adapt to new data. Our study paves the way for lifelong on-device learning and opens up new possibilities for privacy-preserving device personalization.

## 2 Approach

**Preliminaries.**    Neural networks usually need to be quantized to fit the limited memory of edge devices for inference [47, 34]. For a `fp32` linear layer $\mathbf{y}_{fp32} = \mathbf{W}_{fp32}\mathbf{x}_{fp32} + \mathbf{b}_{fp32}$, the `int8`

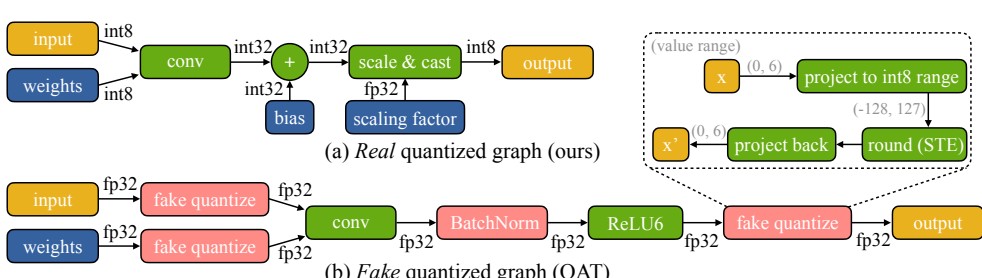

(a) *Real* quantized graph (ours)

(b) *Fake* quantized graph (QAT)

**Figure 2.** *Real* quantized graphs (our optimized graph, designed for *efficiency*) *vs.* *fake* quantized graphs (for QAT, designed for *simulation*). The fake quantize graphs cannot provide memory saving due to floating-point operations. We need to use real quantized graph to fit the tight memory constraint.

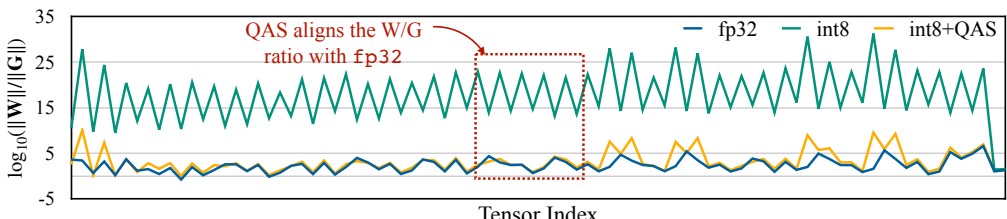

**Figure 3.** The quantized model has a very different weight/gradient norm ratio (*i.e.*, $\|\mathbf{W}\|/\|\mathbf{G}\|$) compared to the floating-point model at training time. QAS stabilizes the $\|\mathbf{W}\|/\|\mathbf{G}\|$ ratio and helps optimization. For example, in the highlighted area, the ratios of the quantized model fluctuate dramatically in a zigzag pattern (weight, bias, weight, bias, ...); after applying QAS, the pattern stabilizes and matches the `fp32` counterpart.

quantized counterpart is:

$$\bar{\mathbf{y}}_{\texttt{int8}} = \texttt{cast2int8}[s_{\texttt{fp32}} \cdot (\bar{\mathbf{W}}_{\texttt{int8}}\bar{\mathbf{x}}_{\texttt{int8}} + \bar{\mathbf{b}}_{\texttt{int32}})], \tag{1}$$

where $\bar{\phantom{x}}$ denotes the tensor being quantized to fixed-point numbers, and $s$ is a floating-point scaling factor to project the results back into `int8` range. We call it *real* quantized graphs (Figure 2(a)) since tensors are in `int8` format. To keep the memory efficiency, we deploy and update the *real* quantized graph on microcontrollers, and keep the updated weights as `int8`. The update formula is: $\bar{\mathbf{W}}'_{\texttt{int8}} = \texttt{cast2int8}(\bar{\mathbf{W}}_{\texttt{int8}} - \alpha \cdot \mathbf{G}_{\bar{\mathbf{W}}})$, where $\alpha$ is the learning rate, and $\mathbf{G}_{\bar{\mathbf{W}}}$ is the gradient of the weights. The gradient computation is also performed in `int8` for better computation efficiency.

We update the real quantized graph for training, which is fundamentally different to quantization-aware training (QAT), where a *fake* quantized graph (Figure 2(b)) is trained on the cloud, and converted to a real one for deployment. As shown in Figure 2(b), the fake quantization graph uses `fp32`, leading to no memory or computation savings. *Real* quantized graphs are for *efficiency*, while *fake* quantized graphs are for *simulation*.

## 2.1 Optimizing Real Quantized Graphs

Unlike fine-tuning floating-point model on the cloud, training with *a real* quantized graph is difficult: the quantized graph has tensors of different bit-precisions (`int8`, `int32`, `fp32`, shown in Equation 1) and lacks Batch Normalization [33] layers (fused), leading to unstable gradient update.

**Gradient scale mismatch.** When optimizing a quantized graph, the accuracy is lower compared to the floating-point counterpart. We hypothesize that the quantization process distorts the gradient update. To verify the idea, we plot the ratio between weight norm and gradient norm (*i.e.*, $\|\mathbf{W}\|/\|\mathbf{G}\|$) for each tensor at the beginning of the training on the CIFAR dataset [40] in Figure 3. The ratio curve is very different after quantization: (1) the ratio is much larger (could be addressed by adjusting the learning rate); (2) the ratio has a different pattern after quantization. Take the highlighted area (red box) as an example, the quantized ratios have a zigzag pattern, differing from the floating-point curve. If we use a fixed learning rate for all the tensors, then the update speed of each tensor would be very different compared to the floating-point case, leading to inferior accuracy. We empirically find that adaptive-learning rate optimizers like Adam [36] cannot fully address the issue (Section 3.2).

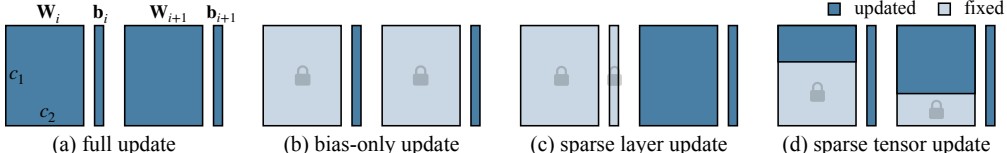

**Figure 4.** Different update paradigms of two linear layers in a deep neural network.

**Quantization-aware scaling (QAS).** To address the problem, we propose a hyper-parameter-free learning rate scaling rule, QAS. Consider a 2D weight matrix of a linear layer $\mathbf{W} \in \mathbb{R}^{c_1 \times c_2}$, where $c_1, c_2$ are the input and output channel. To perform per-tensor quantization*, we compute a scaling rate $s_{\mathbf{W}} \in \mathbb{R}$, such that $\bar{\mathbf{W}}$'s largest magnitude is $2^7 - 1 = 127$:

$$\mathbf{W} = s_{\mathbf{W}} \cdot (\mathbf{W}/s_{\mathbf{W}}) \stackrel{\text{quantize}}{\approx} s_{\mathbf{W}} \cdot \bar{\mathbf{W}}, \quad \mathbf{G}_{\bar{\mathbf{W}}} \approx s_{\mathbf{W}} \cdot \mathbf{G}_{\mathbf{W}}, \tag{2}$$

The process (roughly) preserves the mathematical functionality during the forward (Equation 1), but it distorts the magnitude ratio between the weight and its corresponding gradient:

$$\|\bar{\mathbf{W}}\|/\|\mathbf{G}_{\bar{\mathbf{W}}}\| \approx \|\mathbf{W}/s_{\mathbf{W}}\|/\|s_{\mathbf{W}} \cdot \mathbf{G}_{\mathbf{W}}\| = s_{\mathbf{W}}^{-2} \cdot \|\mathbf{W}\|/\|\mathbf{G}\|. \tag{3}$$

We find that the weight and gradient ratios are off by $s_{\mathbf{W}}^{-2}$, leading to the distorted pattern in Figure 3: (1) the scaling factor is far smaller than 1, making the weight-gradient ratio much larger; (2) weights and biases have different data type (int8 *vs.* int32) and thus have scaling factors of very different magnitude, leading to the zigzag pattern. To solve the issue, we propose Quantization-Aware Scaling (QAS) by compensating the gradient of the quantized graph according to Equation 3:

$$\tilde{\mathbf{G}}_{\bar{\mathbf{W}}} = \mathbf{G}_{\bar{\mathbf{W}}} \cdot s_{\mathbf{W}}^{-2}, \quad \tilde{\mathbf{G}}_{\bar{\mathbf{b}}} = \mathbf{G}_{\bar{\mathbf{b}}} \cdot s_{\mathbf{W}}^{-2} \cdot s_{\mathbf{x}}^{-2} = \mathbf{G}_{\bar{\mathbf{b}}} \cdot s^{-2} \tag{4}$$

where $s_{\mathbf{X}}^{-2}$ is the scaling factor for quantizing input $\mathbf{x}$ (a scalar following [34], note that $s = s_{\mathbf{W}} \cdot s_{\mathbf{x}}$ in Equation 1). We plot the $\|\mathbf{W}\|/\|\mathbf{G}\|$ curve with QAS in Figure 3 (int8+scale). After scaling, the gradient ratios match the floating-point counterpart. QAS enables fully quantized training (int8 for both forward and backward) while matching the accuracy of the floating-point training (Table 1).

## 2.2 Memory-Efficient Sparse Update

Though QAS makes optimizing a quantized model possible, updating the whole model (or even the last several blocks) requires a large amount of memory, which is not affordable for the tinyML setting. We propose to sparsely update the layers and the tensors.

**Sparse layer/tensor update.** Pruning techniques prove to be quite successful for achieving sparsity and reducing model size [29, 30, 48, 31, 50, 49]. Instead of pruning *weights* for inference, we "prune" the *gradient* during backpropagation, and update the model sparsely. Given a tight memory budget, we skip the update of the *less important* parameters to reduce memory usage and computation cost. We consider updating a linear layer $\mathbf{y} = \mathbf{W}\mathbf{x} + \mathbf{b}$ (similar analysis applies to convolutions). Given the output gradient $\mathbf{G}_{\mathbf{y}}$ from the later layer, we can compute the gradient update by $\mathbf{G}_{\mathbf{W}} = f_1(\mathbf{G}_{\mathbf{y}}, \mathbf{x})$ and $\mathbf{G}_{\mathbf{b}} = f_2(\mathbf{G}_{\mathbf{y}})$. Notice that updating the biases does not require saving the intermediate activation $\mathbf{x}$, leading to a lighter memory footprint [12][†]; while updating the weights is more memory-intensive but also more expressive. For hardware like microcontrollers, we also need an extra copy for the updated parameters since the original ones are stored in read-only FLASH [47]. Given the different natures of updating rules, we consider the sparse update rule in three aspects (Figure 4): (1) *Bias update*: how many layers should we backpropagate to and update the biases (bias update is cheap, we always update the biases if we have backpropagated to a layer). (2) *Sparse layer update*: select a subset of layers to update the corresponding weights. (3) *Sparse tensor update*: we further allow updating a subset of weight channels to reduce the cost.

However, finding the right sparse update scheme under a memory budget is challenging due to the large combinational space. For MCUNet [47] model with 43 convolutional layers and weight update ratios from {0, 1/8, 1/4, 1/2, 1}, the combination is about $10^{30}$, making exhaustive search impossible.

---

*For simplicity. We actually used per-channel quantization [34] and the scaling factor is a vector of size $c_2$.
†If we update many layers, the intermediate activation could consume a large memory [18].

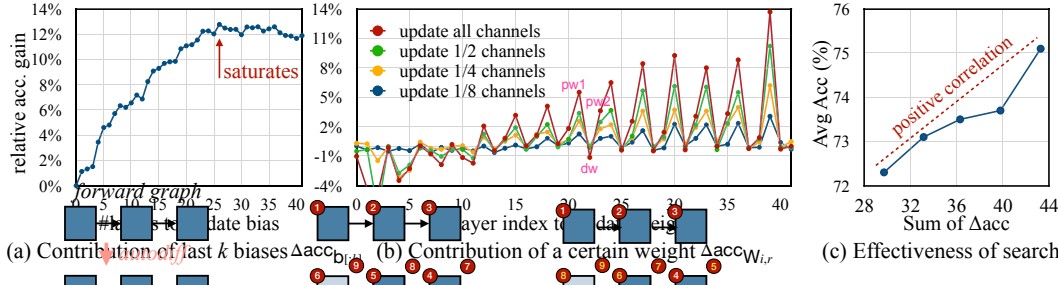

(a) Contribution of last $k$ biases $\Delta\text{acc}_{\mathbf{b}[:k]}$    (b) Contribution of a certain weight $\Delta\text{acc}_{\mathbf{W}_{i,r}}$    (c) Effectiveness of search

**Figure 5.** Contribution analysis of updating biases and weights. **(a)** For bias update, the accuracy generally goes higher as more layers are updated, but plateaus soon. **(b)** For updating the weight of a specific layer, the later layers appear to be more important; the first point-wise conv (pw1) in an inverted bottleneck block [60] appears to be more important; and the gains are bigger with more channels updated. **(c)** The automated selection based on contribution analysis is effective: the actual downstream accuracy shows a positive correlation with $\sum \Delta\text{acc}$.

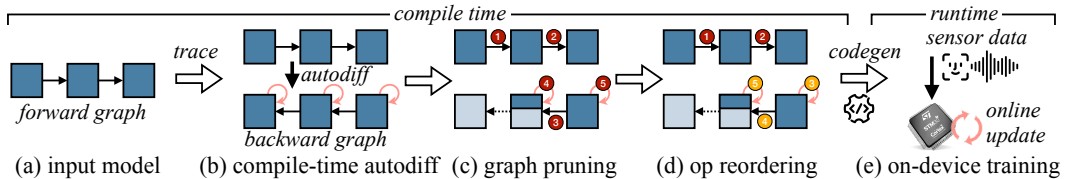

(a) input model    (b) compile-time autodiff    (c) graph pruning    (d) op reordering    (e) on-device training

**Figure 6.** The workflow of our Tiny Training Engine (TTE). **(a,b)** Our engine traces the forward graph for a given model and derives the corresponding backward graph at compile time. The red cycles denote the gradient descent operators. **(c)** To reduce memory requirements, nodes related with frozen weights (colored in light blue) are pruned from backward computation. **(d)** To minimize memory footprint, the gradient descent operators are re-ordered to be interlaced with backward computations (colored in yellow). **(e)** TTE compiles forward and backward graphs using code generation and deploys training on tiny IoT devices (best viewed in colors).

**Automated selection with contribution analysis.** We propose to automatically derive the sparse update scheme by *contribution analysis*. We find the contribution of each parameter (weight/bias) to the downstream accuracy. Given a convolutional neural network with $l$ layers, we measure the accuracy improvement from (1) biases: the improvement of updating *last $k$* biases $\mathbf{b}_l, \mathbf{b}_{l-1}, ..., \mathbf{b}_{l-k+1}$ (bias-only update) compared to only updating the classifier, defined as $\Delta\text{acc}_{\mathbf{b}[:k]}$; (2) weights: the improvement of updating the weight of one extra layer $\mathbf{W}_i$ (with a channel update ratio $r$) compared to bias-only update, defined as $\Delta\text{acc}_{\mathbf{W}i,r}$. An example of the contribution analysis can be found in Figure 5 (MCUNet on Cars [39] dataset; please find more results in appendix Section F). After we find $\Delta\text{acc}_{\mathbf{b}[:k]}$ and $\Delta\text{acc}_{\mathbf{W}i}$ ($1 \leq k, i \leq l$), we solve an optimization problem to find:

$$k^*, \mathbf{i}^*, \mathbf{r}^* = \max_{k,\mathbf{i},\mathbf{r}}(\Delta\text{acc}_{\mathbf{b}[:k]} + \sum_{i\in\mathbf{i}, r\in\mathbf{r}} \Delta\text{acc}_{\mathbf{W}_{i,r}}) \quad \text{s.t. Memory}(k, \mathbf{i}, \mathbf{r}) \leq \text{constraint}, \quad (5)$$

where $\mathbf{i}$ is a collection of layer indices whose weights are updated, and $\mathbf{r}$ is the corresponding update ratios (1/8, 1/4, 1/2, 1). Intuitively, by solving this optimization problem, we find the combination of (#layers for bias update, the subset of weights to update), such that the total contribution are maximized while the memory overhead does not exceed the constraint. The problem can be efficiently solved with evolutionary search (see Section D). Here we assume that the accuracy contribution of each tensor ($\Delta\text{acc}$) can be summed up. Such approximation is quite effective (Figure 5(c)).

## 2.3 Tiny Training Engine (TTE)

The theoretical saving from real quantized training and sparse update does not translate to measured memory saving in existing deep learning frameworks, due to the redundant runtime and the lack of graph pruning. We co-designed an efficient training system, Tiny Training Engine (TTE), to transform the above algorithms into slim binary codes (Figure 6).

**Compile-time differentiation and code generation.** TTE offloads the auto-differentiation from the runtime to the compile-time, generating a static backward graph which can be pruned and optimized (see below) to reduce the memory and computation. TTE is based on code generation: it compiles the

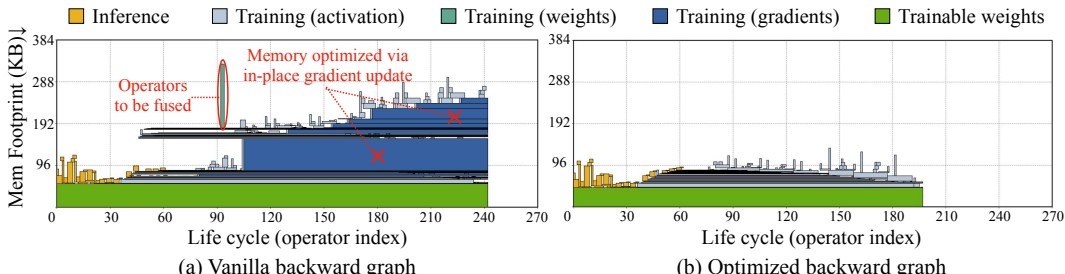

**Figure 7.** Memory footprint reduction by operator reordering. With operator reordering, TTE can apply in-place gradient update and perform operator fusion to avoid large intermediate tensors to reduce memory footprint. We profiled MobileNetV2-w0.35 in this figure (same as Figure 1).

optimized graphs to executable binaries on the target hardware, which minimizes the runtime library size and removes the need for host languages like Python (typically uses Megabytes of memory).

**Backward graph pruning for sparse update.** We prune the redundant nodes in the backward graph before compiling it to binary codes. For sparse layer update, we prune away the gradient nodes of the frozen weights, only keeping the nodes for bias update. Afterwards, we traverse the graph to find unused intermediate nodes due to pruning (*e.g.*, saved input activation) and apply dead-code elimination (DCE) to remove the redundancy. For sparse tensor update, we introduce a *sub-operator slicing* mechanism to split a layer's weights into trainable and frozen parts; the backward graph of the frozen subset is removed. Our compiler translates the sparse update algorithm into measured memory saving, reducing the training memory 7-9× without losing accuracy (Figure 10(a), blue v.s. yellow).

**Operator reordering and in-place update.** The execution order of different operations affects the life cycle of tensors and the overall memory footprint. This has been well-studied for inference [6, 44] but not for training due to the extra complexity. Traditional training frameworks usually derive the gradients of all the trainable parameters before applying the update. Such a practice leads to significant memory waste for storing the gradients. By reordering operators, we can immediately apply the gradient update to a specific tensor (in-place update) before back-propagating to earlier layers, so that the gradient can be released. As such, we trace the dependency of all tensors (weights, gradients, activation) and reorder the operators, so that some operators can be fused to reduce memory footprint (by 2.4-3.2×, Figure 10(a), yellow v.s. red). The memory life cycle analysis in Figure 7 reflects the memory saving from in-place gradient update and operator fusion.

## 3 Experiments

### 3.1 Setups

**Training.** We used three popular tinyML models in our experiments: MobileNetV2 [60] (width multiplier 0.35, backbone 17M MACs, 0.25M Param), ProxylessNAS [13] (width multiplier 0.3, backbone 19M MACs, 0.33M Param), MCUNet [47] (the 5FPS ImageNet model, backbone 23M MACs, 0.48M Param). We pre-trained the models on ImageNet [22] and perform post-training quantization [34]. The quantized models are fine-tuned on downstream datasets to evaluate the transfer learning capacity. We perform the training and memory/latency measurement on a microcontroller STM32F746 (320KB SRAM, 1MB Flash) using a single batch size. To faster obtain the accuracy statistics on multiple downstream datasets, we simulate the training results on GPUs, and we verified that the simulation obtains the same level of accuracy compared to training on microcontrollers. Please refer to the the appendix (Section C) for detailed training hyper-parameters. We also provide a *video demo* of deploying our training system on microcontroller in the appendix (Section A).

**Datasets.** We measure the transfer learning accuracy on multiple downstream datasets and report the average accuracy [37]. We follow [12] to use a set of vision datasets including Cars [39], CIFAR-10 [40], CIFAR-100 [40], CUB [67], Flowers [54], Food [9], and Pets [55][‡]. We fine-tuned the models on all these datasets for 50 epochs following [12]. We also include VWW dataset [20], a

---

[‡]Pets uses CC BY-SA 4.0 license; Cars and ImageNet use the ImageNet license; others are not listed.

**Table 1.** Updating real quantized graphs (`int8`) for the fine-tuning is difficult: the accuracy falls behind the floating-point counterpart (`fp32`), even with adaptive learning rate optimizers like Adam [36] and LARS [68]. QAS helps to bridge the accuracy gap without memory overhead (slightly higher due to randomness). The numbers are for updating the last two blocks of MCUNet-5FPS [47] model.

| Precision | Optimizer | Accuracy (%) (MCUNet backbone: 23M MACs, 0.48M Param ) | | | | | | | | Avg Acc. |
|---|---|---|---|---|---|---|---|---|---|---|
| | | Cars | CF10 | CF100 | CUB | Flowers | Food | Pets | VWW | |
| `fp32` | SGD-M | 56.7 | 86.0 | 63.4 | 56.2 | 88.8 | 67.1 | 79.5 | 88.7 | 73.3 |
| `int8` | SGD-M | 31.2 | 75.4 | 54.5 | 55.1 | 84.5 | 52.5 | 81.0 | 85.4 | 64.9 |
| | Adam [36] | 54.0 | 84.5 | 61.0 | 58.5 | 87.2 | 62.6 | 80.1 | 86.5 | 71.8 |
| | LARS [68] | 5.1 | 64.8 | 39.5 | 9.6 | 28.8 | 46.5 | 39.1 | 85.0 | 39.8 |
| | SGD-M+QAS | 55.2 | 86.9 | 64.6 | 57.8 | 89.1 | 64.4 | 80.9 | 89.3 | **73.5** |

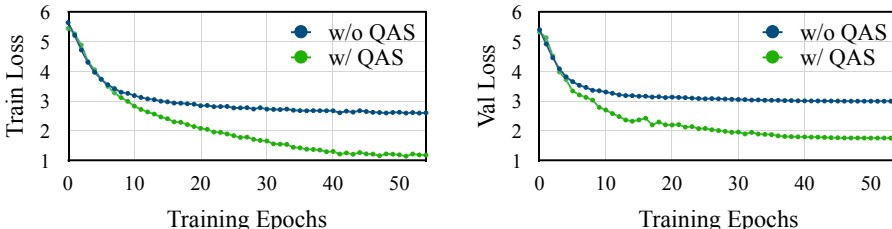

**Figure 8.** Training and validation loss curves w/ and w/o QAS. QAS effectively helps convergence, leading to better accuracy. The results are from updating the last two blocks of the MCUNet model on the Cars dataset.

widely used benchmark for tinyML applications. We train on VWW for 10 epochs following [47]. We used resolution 128 for all datasets and models for a fair comparison.

**Memory estimation.** The memory usage of a computation graph is related to its implementation [6, 44, 47, 46]. We provide two settings for memory measurement: (1) **analytic profiling**: we count the size of *extra* tensors required for backward computation, including the saved intermediate activation, binary truncation task, and the updated weights. The size is implementation-agnostic. It is used for a fast profiling; (2) **on-device profiling**: we measure the actual memory usage when running model training on an STM32F746 MCU (320KB SRAM, 1MB Flash). We used TinyEngineV2 [46] as the backend and 2×2 patch-based inference [46] for the initial stage to reduce the forward peak memory. The *measured* memory determines whether a solution can be deployed on the hardware.

### 3.2 Experimental Results

**Quantization-aware scaling (QAS) addresses the optimization difficulty.** We fine-tuned the last two blocks (simulate low-cost fine-tuning) of MCUNet to various downstream datasets (Table 1). With momentum SGD, the training accuracy of the quantized model (`int8`) falls behind the floating-point counterpart due to the optimization difficulty. Adaptive learning rate optimizers like Adam [36] can improve the accuracy but are still lower than the `fp32` fine-tuning results; it also costs **3×** **memory** consumption due to second-order momentum, which is not desired for tinyML settings. LARS [68] cannot converge well on most datasets despite extensive hyper-parameter tuning (over both learning rate and the "trust coefficient"). We hypothesize that the aggressive gradient scaling rule of LARS makes the training unstable. The accuracy gap is closed when we apply QAS, matching the accuracy of floating-point training at no extra memory cost. The learning curves (fine-tuning) of MCUNet on the Cars dataset w/ and w/o QAS are also provided in Figure 8. Therefore, QAS effectively helps optimization.

**Sparse update obtains better accuracy at lower memory.** We compare the performance of our searched sparse update schemes with two baseline methods: fine-tuning only biases of the last $k$ layers; fine-tuning weights and biases of the last $k$ layers (including fine-tuning the full model, when $k$ equals to the total #layers). For each configuration, we measure the average accuracy on the 8 downstream datasets and the *analytic* extra memory usage. We also compare with a simple baseline by only fine-tuning the classifier. As shown in Figure 9, the accuracy of classifier-only update is low

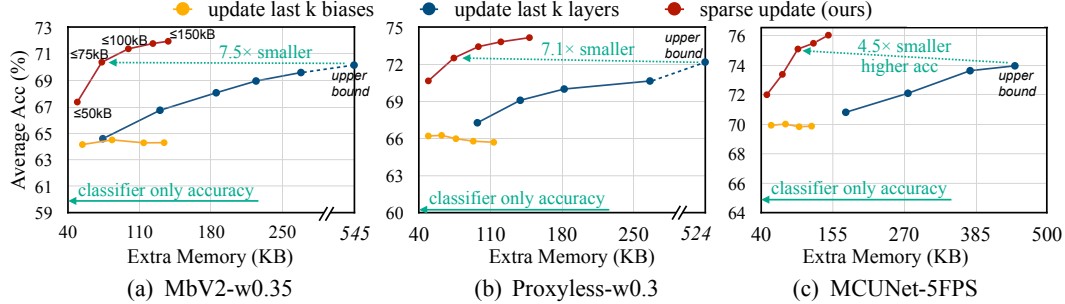

**Figure 9.** Sparse update can achieve higher transfer learning accuracy using 4.5-7.5× smaller extra memory (analytic) compared to updating the last $k$ layers. For classifier-only update, the accuracy is low due to limited capacity. Bias-only update can achieve a higher accuracy but plateaus soon.

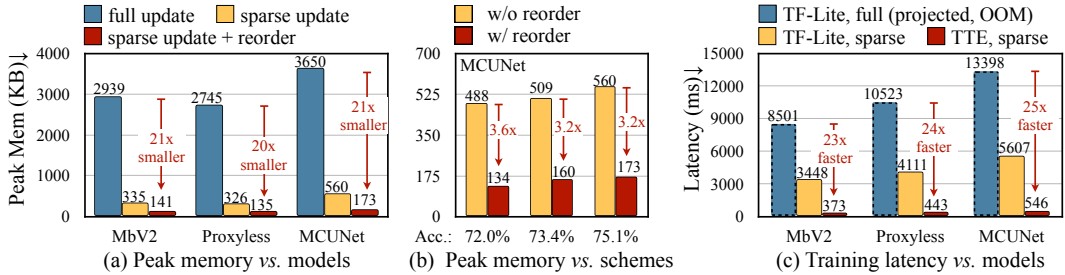

**Figure 10.** *Measured* peak memory and latency: **(a)** Sparse update with TTE graph optimization can reduce the measured peak memory by 20-21× for different models, making training feasible on tiny edge devices. **(b)** Graph optimization consistently reduces the peak memory for different sparse update schemes (denoted by different average transfer learning accuracies). **(c)** Sparse update with TTE operators achieves 23-25× faster training speed compared to the full update with TF-Lite Micro operators, leading to less energy usage. *Note*: for sparse update, we choose the config that achieves the same accuracy as full update.

due to the limited learning capacity. *Updating the classifier alone is not enough; we also need to update the backbone.* Bias-only update outperforms classifier-only update but the accuracy quickly plateaus and does not improve even more biases are tuned. For updating last $k$ layers, the accuracy generally goes higher as more layers are tuned; however, it has a very large memory footprint. Take MCUNet as an example, updating the last two blocks leads to an extra memory surpassing 256KB, making it infeasible for IoT devices/microcontrollers. Our sparse update scheme can achieve higher downstream accuracy at a much lower memory cost: compared to updating last $k$ layers, sparse update can achieve higher downstream accuracy with smaller memory footprint. We also measure the highest accuracy achievable by updating the last $k$ layers (including fine-tuning the full model[§]) as the baseline upper bound (denoted as "upper bound"). Interestingly, our sparse update achieves a better downstream accuracy compared to the baseline best statistics. We hypothesize that the sparse update scheme alleviates over-fitting or makes momentum-free optimization easier.

**Matching cloud training accuracy for tinyML.** Remarkably, the downstream accuracy of our on-device training has *matched or even surpassed* the accuracy of cloud-trained results on tinyML application VWW [20]. Our framework uses 206KB *measured* SRAM while achieving 89.1% top-1 accuracy for on-device training (we used gradient accumulation for the VWW dataset; see the appendix Section C for details). The result is higher than the accuracy of the same model reported by the state-of-the-art solution MCUNet (88.7%, trained on cloud and deployed to MCU). Both settings transfer the ImageNet pre-trained model to VWW. The on-device accuracy is far above the common requirement for tinyML (>80% by MLPerf Tiny [8]) and surpassed the results of industry solution TF-Lite Micro+MobileNetV2 (86.2% [47] under 256KB, *inference-only, no training support*).

**Tiny Training Engine: memory saving.** We measure the training memory of three models on STM32F746 MCU to compare the memory saving from TTE. We measure the peak SRAM usage

---

[§]Note that fine-tuning the entire model does not always lead to the best accuracy. We grid search for the best $k$ on Cars dataset: $k = 36$ for MobileNetV2, 39 for ProxylessNAS, 12 for MCUNet, and apply it to all datasets.

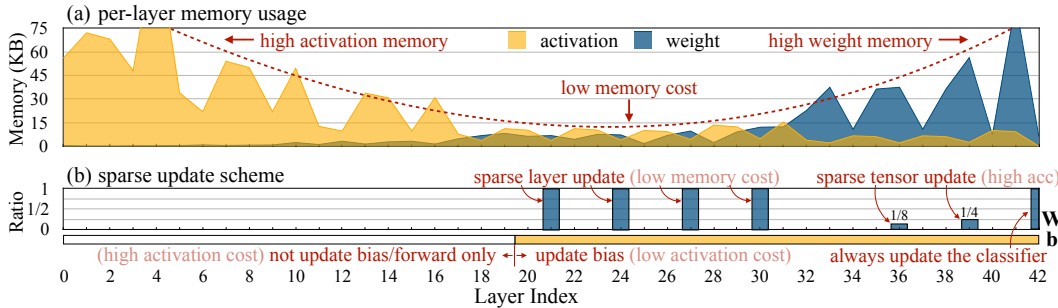

**Figure 11. (a)** The weight and activation memory cost of updating *each* layer of MCUNet (analytic). We find that the activation cost is high for the starting layers; the weight cost is high for the later layers; the overall memory cost is low for the middle layers. **(b)** Dissecting the sparse update scheme: we update the biases of the last 22 layers due to its low activation cost. For weight update, we update some middle layers due to its low memory cost, and update partial channels of the two later layers since they are important for accuracy (Figure 5).

under three settings: general full update, sparse update, and sparse update with TTE graph reordering (Figure 10(a)). The sparse update effectively reduces peak memory by 7-9× compared to the full update thanks to the graph pruning mechanism, while achieving the same or higher transfer learning accuracy (compare the data points connected by arrows in Figure 9). The memory is further reduced with operator reordering, leading to 20-21× total memory saving. With both techniques, the training of all 3 models fits 256KB SRAM. We also compare the memory saving of reordering under different update schemes on MCUNet (Figure 9(b), indicated by different accuracy levels). Reordering consistently reduces the peak memory for different sparse update schemes of varying learning capacities.

**Tiny Training Engine: faster training.** We further measure the training latency per image on the STM32F746 MCU with three settings: full update with TF-Lite Micro kernels, sparse update with TF-Lite Micro kernels, and sparse update with TTE kernels (Figure 10(c)). Notice that TF-Lite *does not* support training; we just used the kernel implementation to measure latency. By graph optimization and exploiting multiple compiler optimization approaches (such as loop unrolling and tiling), our sparse update + TTE kernels can significantly enhance the training speed by 23-25× compared to the full update + TF-Lite Micro kernels, leading to energy saving and making training practical. Note that TF-Lite with full update leads to OOM, so we report the projected latency according to the average speed of each op type (marked in dashed columns).

### 3.3 Ablation Studies and Analysis

**Dissecting update schedules.** We visualize the update schedule of the MCUNet [47] model searched under 100KB extra memory (analytic) in Figure 11 (lower subfigure (b), with 10 classes). It updates the biases of the last 22 layers, and sparsely updates the weights of 6 layers (some are sub-tensor update). The initial 20 layers are frozen and run forward only. To understand why this scheme makes sense, we also plot the memory cost from activation and weight when updating *each* layer in the upper subfigure (a). We see a clear pattern: the activation cost is high for the initial layers; the weight cost is high for the ending layers; while the total memory cost is low when we update the middle layers (layer index 18-30). The update scheme matches the memory pattern: to skip the initial stage of high activation memory, we only update biases of the later stage of the network; we update the weights of 4 intermediate layers due to low overall memory cost; we also update the partial weights of two later layers (1/8 and 1/4 weights) due to their high contribution to the downstream accuracy (Figure 5). Interestingly, all the updated weights are from the first point-wise convolution in each inverted residual block [60] as they generally have a higher contribution to accuracy (the peak points on the zigzag curve in Figure 5(b)).

**Effectiveness of contribution analysis.** We verify if the update scheme search based on contribution analysis is effective. We collect several data points during the search process (the update scheme and the search criteria, *i.e.*, the sum of Δacc). We train the model with each update scheme to get the average accuracy on the downstream datasets (the real optimization target) and plot the comparison in Figure 5(c). We observe a positive correlation, indicating the effectiveness of the search.

**Sub-channel selection.** Similar to weight pruning, we need to select the subset of channels for sub-tensor update. We update the last two blocks of the MCUNet [47] model and only 1/4 of the weights for each layer to compare the accuracy of different channel selection methods (larger magnitude, smaller magnitude, and random). The results are quite similar (within 0.2% accuracy difference). Channel selection is not very important for transfer learning (unlike pruning). We choose to update the channels with a larger weight magnitude since it has slightly higher accuracy.

## 4 Related Work

**Efficient transfer learning.** There are several ways to reduce the transfer learning cost compared to fine-tuning the full model [38, 21, 37]. The most straightforward way is to only update the classifier layer [15, 23, 26, 61], but the accuracy is low when the domain shift is large [12]. Later studies investigate other tuning methods including updating biases [12, 70], updating normalization layer parameters [53, 25], updating small parallel branches [12, 32], *etc*. These methods only reduce the trainable parameter number but lack the study on system co-design to achieve real memory savings. Most of them do not fit tinyML settings (cannot handle quantized graph and lack of BatchNorm [33]).

**Systems for deep learning.** The success of deep learning is built on top of popular training frameworks such as PyTorch [56], TensorFlow [5], MXNet [16], JAX [10], *etc*. These systems usually depend on a host language (*e.g*. Python) and various runtimes, which brings significant overhead (>300MB) and does not fit tiny edge devices. Inference libraries like TVM [17], TF-Lite [3], MNN [35], NCNN [1], TensorRT [2], and OpenVino [65] provide lightweight runtime environments but do not support training (only MNN has preliminary support for full model training). None of the existing frameworks can fit tiny IoT devices with tight memory constraints.

**Tiny deep learning on microcontrollers.** Tiny deep learning on microcontrollers is challenging. Existing work explores model compression (pruning [29, 30, 48, 31, 50, 69, 45], quantization [29, 57, 66, 19, 59, 42, 47, 34]) and neural architecture search [71, 72, 64, 47, 7, 43, 24, 51, 47, 46] to reduce the required resource of deep learning models. There are several deep learning systems for tinyML (TF-Micro [5], CMSIS-NN [41], TinyEngine [47], MicroTVM [17], CMix-NN [14], *etc*.). However, the above algorithms and systems are only for inference but not training. There are several preliminary attempts to explore training on microcontrollers [58, 28, 63, 62]. However, due to the lack of efficient algorithm and system support, they are only able to tune one layer or a very small model, while our work supports the tuning of modern CNNs for real-life applications.

## 5 Conclusion

In this paper, we propose the first solution to enable tiny on-device training on microcontrollers under a tight memory budget of 256KB. Our algorithm system co-design solution significantly reduces the training memory (more than 1000× compared with PyTorch and TensorFlow) and per-iteration latency (more than 20× speedup over TensorFlow-Lite Micro), allowing us to obtain higher downstream accuracy. Our study suggests that tiny IoT devices can not only perform inference but also continuously adapt to new data for lifelong learning.

**Limitations and societal impacts.** Our work achieves the first practical solution for transfer learning on tiny microcontrollers. However, our current study is limited to vision recognition with CNNs. In the future, we would like to extend to more modalities (*e.g*., audio) and more models (*e.g*., RNNs, Transformers). Our study improves tiny on-device learning, which helps to protect the privacy on sensitive data (*e.g*., healthcare). However, to design and benchmark our method, we experimented on many downstream datasets, leading to a fair amount of electricity consumption.

## Acknowledgments

We thank National Science Foundation (NSF), MIT-IBM Watson AI Lab, MIT AI Hardware Program, Amazon, Intel, Qualcomm, Ford, Google for supporting this research.

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
