# Supplementary Material
# On-Device Training Under 256KB Memory

**Contents**

## A    Video Demo

We prepared a video demo showing that we can deploy our framework to a microcontroller (STM32F746, 320KB SRAM, 1MB Flash) to enable on-device learning. We adapt the MCUNet model (pre-trained on ImageNet) to classify whether there is a person in front of the camera or not. The training leads to decent accuracy within the tight memory budget. Please find the demo here: https://youtu.be/XaDCO8YtmBw.

The training is performed with 100 sample images from the VWW dataset [2] fed through the camera (50 positive and 50 negative). The total (pure) training throughput for the pipeline (including overheads like camera IO) is shown in the Figure. S1. The total training time would be around minutes. This is quite affordable for tiny on-device learning applications.

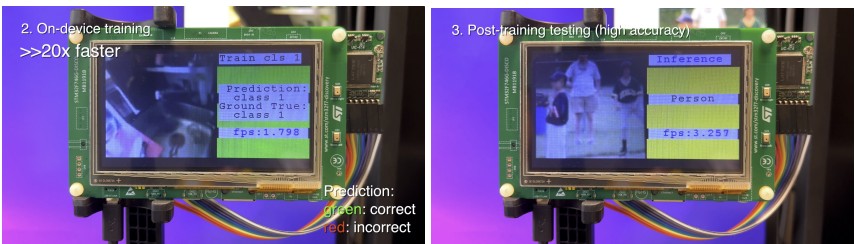

**Figure S1.** A screenshot of our video demo.

## B    Variance of Different Runs

We notice that the variance of different runs is quite small in our experiments. Here we provide detailed information about the variance.

Firstly, if we use the same random seed for the data loader, we will get *exactly the same* results for multiple runs. The weight quantization process after each iteration (almost) eliminates the non-determinism from GPU training*. Therefore, we study the randomness from different random seeds in *data shuffling*. Here we provide the results of 3 runs in Table S1 to show the variance. We train the MobileNetV2-w0.35 model with the sparse update scheme (searched under 100KB analytic memory usage) 3 times independently. We find the variance is very small, especially when we report the average accuracy (for most of our results): the standard derivation is only ±0.07%.

**Table S1.** The variance between different runs is small, especially when we report the average accuracy (only ±0.07%). Results obtained by training MobileNetV2-w0.35 for three times using the sparse update scheme searched under 100KB analytic memory constraint.

| Runs | Accuracy (%) | | | | | | | | Avg Acc. |
|------|------|------|-------|------|---------|------|------|------|------|
|      | Cars | CF10 | CF100 | CUB  | Flowers | Food | Pets | VWW  |      |
| run1 | 51.59 | 87.03 | 63.89 | 54.14 | 85.95 | 62.28 | 77.84 | 88.34 | 71.38 |
| run2 | 52.87 | 86.8  | 63.81 | 54.87 | 85.30 | 62.45 | 77.30 | 88.65 | 71.50 |
| run3 | 52.49 | 87.13 | 63.80 | 55.16 | 85.35 | 61.99 | 77.08 | 88.21 | 71.40 |
| mean | 52.32 | 86.99 | 63.83 | 54.72 | 85.53 | 62.24 | 77.41 | 88.40 | **71.43** |
| ±std | ±0.66 | ±0.17 | ±0.05 | ±0.52 | ±0.36 | ±0.23 | ±0.39 | ±0.22 | **±0.07** |

## C    Training Setups & Discussions

In this section, we introduce detailed training setups and discuss the reasons that lead to several design choices.

We used SGD optimizer+QAS for training. We set weight decay as 0 since we observed no overfitting during experiments. This is also a common choice in transfer learning [6]. We find the initial

---

*https://developer.download.nvidia.com/video/gputechconf/gtc/2019/presentation/s9911-determinism-in-deep-learning.pdf

learning rate significantly affects the accuracy, so we extensively tuned the learning rate for each run to report the best accuracy. We used cosine learning rate decay and performed warm-up [4] for 1 epoch on VWW and 5 epochs on other datasets. We used Ray [9] for experiment launching and hyper-parameter tuning.

**Data type of the classifier.**  During transfer learning, we usually need to randomly initialize the classifiers (or add some classes) for novel categories. Although the backbone is fully quantized for efficiency, we find that using a floating-point classifier is essential for transfer learning performance. Using a floating-point classifier is also cost-economical since the classifier consists of a very small part of the model size (0.3% for 10 classes).

We compare the results of the quantized classifier and floating-point classifier in Table S2. We update the last two blocks of the MCUNet model with SGD-M optimizer and QAS to measure the downstream accuracy. We find that keeping the classifier as floating-point significantly improves the downstream accuracy by 2.3% (on average) at a marginal overhead. *Therefore, we use floating-point for the classifier by default.*

**Table S2.** Keeping the classifier as floating-point greatly improves the downstream accuracy.

| fp32 classifier | Accuracy (%) | | | | | | | | Avg Acc. |
|---|---|---|---|---|---|---|---|---|---|
| | Cars | CF10 | CF100 | CUB | Flowers | Food | Pets | VWW | |
| ✗ | 50.8 | 86.1 | 62.7 | 56.8 | 82.5 | 61.7 | 80.8 | 87.8 | 71.2 |
| ✓ | 55.2 | 86.9 | 64.6 | 57.8 | 89.1 | 64.4 | 80.9 | 89.3 | **73.5** |

**Single-batch training & momentum.**  For on-device training on microcontrollers, we can only fit batch size 1 due to the tight memory constraint. However, single-batch training has very low efficiency when simulated on GPUs since it cannot leverage the hardware parallelism, making experiments slow. We study the performance gap between single-batch training and normal-batch training (batch size 128) to see if we can use the latter as an approximation.

We compare the results of different batch sizes in Table S3, with and without momentum. Due to the extremely low efficiency of single-batch training, we only report results on datasets of a smaller size. We used SGD+QAS as the optimizer and updated the last two blocks of the MCUNet [8] model. We extensively tuned the initial learning rate to report the best results.

**Table S3.** Momentum helps transfer learning with batch size 128, but not with batch size 1; without momentum, we can use the normal-batch training results as an approximation for single-batch training. Results obtained by updating the last two blocks of MCUNet [8] with SGD+QARS.

| Batch size | Momentum | Mem Cost | Accuracy (%) | | | | | Avg Acc. |
|---|---|---|---|---|---|---|---|---|
| | | | Cars | CUB | Flowers | Pets | VWW | |
| 128 (GPU simulate) | 0.9 | 2× | 55.2 | 57.8 | 89.1 | 80.9 | 89.3 | 74.4 |
| | 0 | 1× | 47.8 | 57.2 | 87.3 | 80.8 | 88.8 | 72.4 |
| 1 (tinyML) | 0.9 | 2× | 47.8 | 54.8 | 88.5 | 80.5 | 86.2 | 71.5 |
| | 0 | 1× | 51.1 | 56.2 | 88.7 | 79.3 | 86.0 | 72.3 |

We can make two observations:

1. Firstly, momentum helps optimization for normal-batch training as expected (average accuracy 74.4% *vs.* 72.4%). However, it actually makes the accuracy slightly worse for the single-batch setting (71.5% *vs.* 72.3%). Since using momentum will double the memory requirement for updating parameters (assume we can safely quantize momentum buffer; otherwise the memory usage will be 5× larger), we will not use momentum for tinyML on-device learning.

2. Without momentum, normal-batch training, and single-batch training lead to a similar average accuracy (72.4% *vs.* 72.3%), allowing us to use batched training results for evaluation.

Given the above observation, *we report the results of batched training without momentum by default*, unless otherwise stated.

**Gradient accumulation.** With the above training setting, we can get a similar average accuracy compared to actual on-device training on microcontrollers. The reported accuracy on each dataset is quite close to the real on-device accuracy, with *only one exception*: the VWW dataset, where the accuracy is 2.5% lower. This is because VWW only has two categories (binary classification), so the information from each label is small, leading to unstable gradients. For the cases where the number of categories is small, we can add gradient accumulation to make the update more stable. We show the comparison of adapting the pre-trained MCUNet model in Table S4. The practice closes the accuracy gap at a small extra memory cost (11%), allowing us to get 89.1% top-1 accuracy within 256KB memory usage.

To provide a clear comparison, we *do not* apply gradient accumulation in our experiments except for this comparison.

**Table S4.** Gradient accumulation helps the optimization on datasets with a small category number. Numbers obtained by training with batch size 1, the same setting as on microcontrollers.

| model | accumulate grad | SRAM | VWW accuracy |
|---|---|---|---|
| MCUNet-5FPS | ✗ | 160KB | 86.6% |
| | ✓ | 188KB | 89.1% |

## D  Evolutionary Search *vs.* Random Search

We find that evolutionary search can efficiently explore the search space to find a good sparse update scheme given a memory constraint. Here we provide the comparison between evolutionary search and random search in Figure S2. We collect the curves when searching for an update scheme of the MCUNet-5FPS [8] model under 100KB memory constraint (analytic). We find that evolutionary search has a much better sample efficiency and can find a better final solution (higher sum of $\Delta$acc) compared to random search. The search process is quite efficient: we can search for a sparse update scheme within 10 minutes based on the contribution information. Note that we use the *same* update scheme for all downstream datasets.

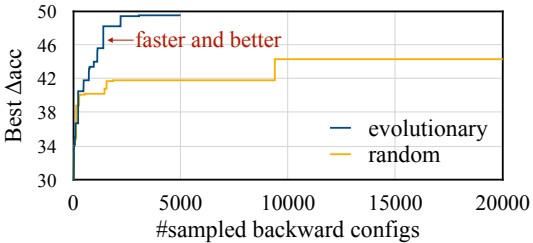

**Figure S2.** Evolutionary search has a better sample efficiency and leads to a better final result compared with random search when optimizing sparse update schemes.

## E  Amount of Compute

To evaluate the performance of different training schemes, we simulate the training on GPUs to measure the average accuracy on 8 downstream datasets. Thanks to the small model size (for the tinyML setting) and the small dataset size, the training cost for each scheme is quite modest: it only takes **3.2 GPU hours** for training on all 8 downstream datasets (cost for one run; do not consider hyper-parameter tuning).

For the pre-training on ImageNet [3], it takes about **31.5 GPU hours** (300 epochs). Note that we only need to pre-train each model *once*.

We performed training with NVIDIA GeForce RTX 3090 GPUs.

# F    More Contribution Analysis Results

Here we provide the contribution analysis results of the MobileNetV2-w0.35 [10] and ProxylessNAS-w0.3 [1] on the Cars dataset [7] (Figure S3 and S4). The pattern is similar to the one from the MCUNet model: the later layers contribute to the accuracy improvement more; within each block, the first point-wise convolutional layer contributes to the accuracy improvement the most.

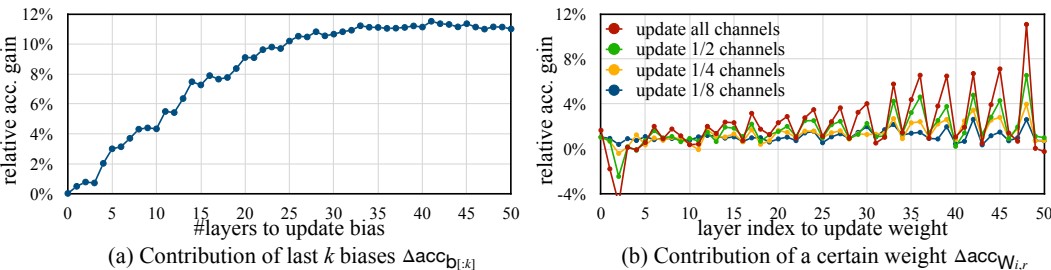

(a) Contribution of last $k$ biases $\Delta\mathrm{acc}_{b_{[:k]}}$  (b) Contribution of a certain weight $\Delta\mathrm{acc}_{W_{i,r}}$

**Figure S3.** Contribution analysis of updating biases and weights for MobileNetV2-w0.35 [10].

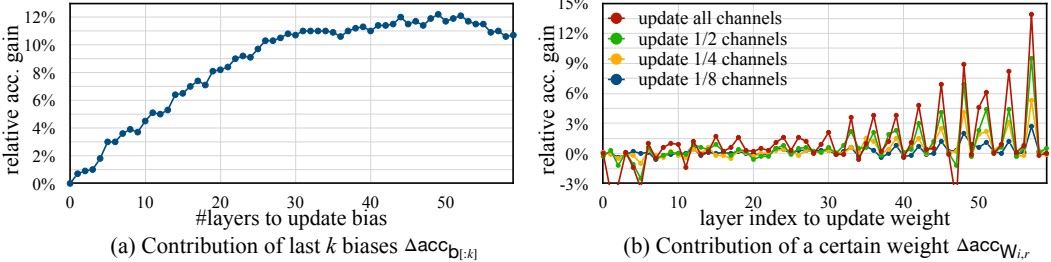

(a) Contribution of last $k$ biases $\Delta\mathrm{acc}_{b_{[:k]}}$  (b) Contribution of a certain weight $\Delta\mathrm{acc}_{W_{i,r}}$

**Figure S4.** Contribution analysis of updating biases and weights for ProxylessNAS-w0.3 [1].

# G Other Partial Update Methods That Did Not Work

During our experiments, we also considered other efficient partial update methods (apart from sparse layer/tensor update) but they did not work well. Here are a few methods we tried but failed:

**1. Low-rank update.** LoRA [5] aims to adapt a model by adding a low-rank decomposed weight to each of the original weight matrix. It is designed for adapting large language models, but could potentially be applied here. Specifically, LoRA freezes the original weight $\mathbf{W} \in \mathbb{R}^{c \times c}$ but trains a small $\Delta \mathbf{W} = \mathbf{MN}$, where $\mathbf{M} \in \mathbb{R}^{c \times c'}, \mathbf{N} \in \mathbb{R}^{c' \times c}, c' << c$. The low-rank decomposed $\Delta \mathbf{W}$ has much fewer parameters compared to $\mathbf{W}$. After training, we can merge $\Delta \mathbf{W}$ so that no extra computation is incurred: $\mathbf{y} = (\mathbf{W} + \Delta \mathbf{W})\mathbf{x}$. However, such method does not work in our case:

1. The weights are quantized in our models. If we merge $\Delta \mathbf{W}$ and $\mathbf{W}$, we will produce a new weight $\mathbf{W}' = \Delta \mathbf{W} + \mathbf{W}$ that has the same size as $\mathbf{W}$, taking up a large space on the SRAM (that is why we need the sparse tensor update).

2. Even if we can tolerate the extra memory overhead by running $\mathbf{y} = \mathbf{Wx} + \Delta \mathbf{Wx}$, the $\Delta \mathbf{W}$ is randomly initialized and we empirically find that it is difficult to update a quantized weight from scratch, leading to worse performance.

**2. Replacing convolutions with lighter alternatives.** As shown in the contribution curves (Figure 4 in the main paper, Figure S3, and Figure S4), the first point-wise convolutional layer in each block has the highest contribution to accuracy. We tried replacing the first point-wise convolutional layer with a lighter alternative, like grouped convolutions. However, although such replacement greatly reduces the cost to update the layers, it also hinders transfer learning accuracy significantly. Therefore, we did not choose to use such modification. It also involves extra complexity by changing model architectures, which is not desired.