# OpenReview forum: "On-Device Training Under 256KB Memory"
_NeurIPS.cc/2022/Conference — NeurIPS 2022 Accept_

### Official Review · Reviewer_xaNn · 2022-07-06

**Rating:** 6
**Confidence:** 5
**Soundness:** 2 fair
**Presentation:** 3 good
**Contribution:** 3 good

**Summary:**

This paper considers a timely topic, especially with the explosion of tiny devices with limited computational resources. It is a trend to deploy the lifelong learning tasks directly on the devices and realize user customization. The authors propose the algorithm-system co-design to fit the rigorous setting, even with only 256KB. The model adaptation methods for gradient calibration and sparse update are straightforward but experimentally work well. This framework matches the practical demands in an edge environment, and thus is a meaningful solution for machine learning analysis on IoT and MCUs. The experiments take commodity on-device learning settings and present promising performance in terms of model size, memory cost, and accuracy.

**Questions:**

Please see the four questions in the weakness part.

**Limitations:**

None.

**Strengths And Weaknesses:**

Generally, this is an interesting paper with a clear methodology design and performance evaluation. However, I have some concerns (about the weakness) as follows.

1. The effectiveness of the proposed model compression and learning strategy are highly related to the processing pattern of the underlying hardware. For example, not all the commodity processing chips can support the specific int8 operations (or efficiently handle these operations). The true acceleration of sparse computing relies on specific chips. So I am interested in whether the proposed methods can generalize to other commodity devices (e.g., extending to the NVIDIA Jetson or a mobile phone’s neural chips, not just the specific STM series).

2. The experiments are mainly based on classification and adopt transfer learning with limited epochs on downstream tasks. What about the performance of segmentation and detection tasks?

3. Also, supporting long-time (a large number of epochs) training is critical for on-device lifelong learning. Can this method train from scratch, instead of fine-tuning the pre-trained models?

4. Handling the learning procedure timely is more critical. I think it is better to present the performance of the training/inference speed or time cost when applying to the downstream tasks.

---

> ### Author Response · Authors · 2022-08-02
> **Author Response**
>
> We thank reviewer xaNn’s useful comments and would like to respond as follows:
>
> **Q1: Can the proposed methods generalize to other commodity devices (e.g., NVIDIA Jetson or Neural chips, not just STM series)? Does the sparse update rely on specific chips to achieve true speedup?**
>
> - **The int8 operations are well-supported on commodity devices**: The int8 operations used in our method follow a standard TF-Lite protocol [33]. It is not specific to the STM series; instead, it is widely supported on a wide range of hardware like NVIDIA GPUs (CUDA Cores, Tensor Cores), Arm CPUs, Intel’s CPUs, Qualcomm’s SoCs, etc. Therefore, our quantized training method is general and can be extended to different devices.
> - **Our techniques can bring general speedup**: The other components like sparse update, compile-time differentiation, backward graph pruning, operator reordering, etc. are not specific to any device or platform. They are general techniques to make on-device adaptation more efficient and do not require special support from the underlying hardware.
> - **Speedup from the sparse update on more edge platforms**: Our sparse update uses channel/op level sparsity and does **not** involve fine-grained sparsity. Therefore it can accelerate training on various hardware platforms. We report the latency (ms) of training MobilenetV2-0.35 with Tiny Training Engie (TTE) using input size 1x3x128x128 in the following table. Our sparse update scheme shows consistent speedup (1.4x-3.0x) over conventional dense/full update while maintaining the same accuracy.
>
> |        | Dense-Update | Sparse Update (ours) | Speedup |
> |---------------------|:------------:|----------------------|:-------:|
> |   Raspberry Pi CPU  |    74.0ms    |         24.4ms        |  3.0x  |
> | Qualcomm S8Gen1 CPU |    39.9ms    |         28.4ms        |  1.4x  |
> |   Jetson Nano GPU   |     5.2ms     |         2.8ms         |  1.8x  |
>
> **Q2: The performance of segmentation and detection tasks.**
>
> TinyML suffers from an extremely small memory size. It is very difficult to support even the **inference** of applications like segmentation and object detection [47] because of the large input resolution. For example, a single 640x640x3 image (a widely used resolution for the MSCOCO dataset) takes 1200 kB to store and already exceeds the available memory on a microcontroller (320KB), let alone the cost for neural networks’ forward and backward. Segmentation and detection need special designs to handle the large resolution challenge, and we leave it for future work.
>
> **Q3: Can this method train from scratch instead of fine-tuning the pre-trained models?**
>
> Our method can also support training from scratch. Experiments in supplementary Section K show that our QAS (with pre-measured weight and activation ranges) is still effective for these non-transfer learning scenarios: training with QAS consistently improves the average accuracy on 8 datasets from 48.3% to 65.4%. Please refer to Section K for details.
>
>  |Method  | Average of Other Datasets |  Cars     |CF10      |CF100      |CUB    |Flowers      |Foods | Pets    |VWW |
> |---------------------|:------------:|:------------:|:------------:|:------------:|:------------:|:------------:|:------------:|:------------:|:------------:|
> | SGD-M   | 48.3 | 15.8   | 63.5   | 41.0   |30.3    |69.8     | 34.1     |52.5      | 79.2      |
> | SGD-M+QAS  | 65.4 | 52.0    |85.5     |58.7      |47.1   |78.2    |53.1     |63.3      |85.6   |
>
>
>
>
>
> We use pre-training because it usually delivers better performance on downstream tasks compared to training from scratch. Recent lifelong learning work [a, b] also performed fine-tuning from pre-trained weights rather than training from scratch.
>
> [a] Mehta et al., An Empirical Investigation of the Role of Pre-training in Lifelong Learning
>
> [b] Wang et al., Learning to Prompt for Continual Learning
>
> **Q4: Show the training speed or time cost when applying to the downstream tasks.**
>
> We have provided the training latency in Figure 9(c). Compared with baselines, our method achieves more than 20x training speed up. For example, it takes 325ms to update MobileNetV2-w0.35 on one image, which is 29x faster than the existing baseline (9456ms). So if a new sample comes in, we can finish the training within a second for a timely update.
>
> For tinyML applications, we usually need a small number of samples for training, so the total training time is also modest. For example, we have built a real demo of fine-tuning the model for a new task (Visual Wake Words) with 100 training images in supplementary Section A. The total training time for MobileNetV2-w0.35 would be just 33s, well within a minute. Given the modest total training time, we believe our framework can enable a vast range of realistic low-power on-device training applications.

---

> > ### Comment · Reviewer_xaNn · 2022-08-08
> > **Author Rebuttal Acknowledgement**
> >
> > Thanks for the author's response. You have addressed my concerns.

---

### Official Review · Reviewer_HCuR · 2022-07-09

**Rating:** 8
**Confidence:** 3
**Soundness:** 4 excellent
**Presentation:** 4 excellent
**Contribution:** 4 excellent

**Summary:**

This paper proposes a novel and efficient framework for the on-device training with 256KB memory. In practice, this paper introduces Quantization-Aware Scaling (QAS) technique to stabilize the quantized training with mixed-bit precision, and present the sparse update technique to save the memory footprint. Besides, the Tiny Training Engine (TTE) is deigned to implement the proposed methods in an MCU system.

**Questions:**

1. Could you show the total training time on the device for three models on different datasets, and show the comparison with GPU. If the training time is too long, the application of these methods may be limited.
2. Can you explain the necessity of on-device training technique and its application scenarios? Training on the cloud device and updating the parameters on the IoT device seems to be more promising.
3. Whether the on-device training has strict requirements on the structure of the neural network models. The plain network seems to be more memory-saving, but this paper do not adopt it. The datasets and models used in the paper are small, some tricks for large models and large datasets seems to be not such important to accuracy, like shortcut connection, attention block, swish activation function.

**Limitations:**

Yes.

**Strengths And Weaknesses:**

Strengths
1. This paper is well-written and easy to understand.
2. This paper propose a complete hardware-software co-design scheme for training on the MCU devices.
3. The experiments and analysis are sufficient to support the proposed methods, the experimental setup is detailed.

Weaknesses
1. Even with much optimization, the inference speed is very slow on the MCU device, is training really necessary and suitable?

---

> ### Author Response · Authors · 2022-08-02
> **Author Response**
>
> We thank the reviewer very much for the positive comments and insightful suggestions.
>
> **Q1: The necessity of on-device training and its applications. Compare training on the cloud and update the parameters locally.**
>
> On-device training has a broad range of applications: it can customize the deep learning models for personalized use cases. For example, Google uses on-device learning to improve the smart keyboard prediction; voice assistants like Alexa and Siri can adapt to different users’ accents; cameras can continually recognize more objects to facilitate smart home and smart manufacturing.
>
> On-device training also has unique advantages. Firstly, it can protect users’ privacy by keeping data locally, especially for sensitive scenarios like healthcare. Secondly, it reduces the data transfer cost and cloud operating cost (which could be huge considering the billions of IoT devices in our daily life). Thirdly, it enables new learning paradigms like life-long learning and federated learning.
>
> On the contrary, uploading the data to the cloud for training can lead to serious privacy and security issues. It may not be feasible as the regularizations become more strict (e.g., the EU’s GDPR makes it very difficult to collect user data in a centralized cloud for training). It also leads to higher cloud operating costs. Therefore, we believe efficient on-device learning has unique advantages and will enable various novel applications.
>
> **Q2: Is the training time reasonable for on-device learning applications?**
>
> Our framework can enable practical on-device training applications even on a small microcontroller with limited computation capacity. We have built a video demo (see [here]([https://drive.google.com/drive/folders/1E0MU7rPHy7NbnlZDKVAIGuN1TTwogF9U?usp=sharing](https://drive.google.com/drive/folders/1E0MU7rPHy7NbnlZDKVAIGuN1TTwogF9U?usp=sharing)) and the supplementary Section A) showing that we can transfer a model to a new target task (Visual Wake Words) within only 200 seconds (which could be further reduced by 3x if we make a good synchronization).
>
> For the detailed timing breakdown of finetuning MobileNetV2-w0.35, we need 325ms to train on one sample. For tinyML applications, we usually need a small number of samples (e.g., <100) to recognize an object, so the total training time is quite acceptable (within minutes). For lifelong learning settings, the training on streaming new data is amortized during a very long period, so the training cost for a certain period is still very affordable.
>
> We also compare the training time for the same workload with a cloud GPU NVIDIA GeForce RTX 3090 using PyTorch, and it takes 20.4ms to train on one sample, which is an order of magnitude faster than our tinyML setting. However, the GPU power is 350W, **three orders of magnitude** higher than microcontrollers (360mW), which is not suitable for tinyML. Given the modest total training time analyzed above, we believe our framework can enable a vast range of realistic low-power on-device training applications.
>
> **Q3: The choice of the used neural network structures.**
>
> Our proposed techniques do not rely on a specific model architecture. They are **general techniques** and can be applied to different CNN model architectures. We chose the three network architectures since they are widely used vision models in tinyML settings. For example, the MCUNet model [45] achieves state-of-the-art accuracy for tinyML applications, outperforming other network architectures in terms of accuracy vs. memory tradeoff. Note that to be friendly for quantization and deployment, the three models do not have attention blocks or Swish activation functions.

---

> > ### Comment · Reviewer_HCuR · 2022-08-08
> > **Thanks for the response**
> >
> > I have read all the reviews and author response, the authors made significant efforts to address all the raised concerns. The demo videos are interesting, and training on MCU seems to be a promising technique for AI application. I would keep my decision as strong accept.

---

### Official Review · Reviewer_JJZV · 2022-07-11

**Rating:** 6
**Confidence:** 5
**Soundness:** 3 good
**Presentation:** 4 excellent
**Contribution:** 3 good

**Summary:**

This paper presents a system-algorithm co-design approach towards conducting transfer learning training on tiny microcontroller-based systems with less than 256KB memory. The ideas focus on: (1) improving optimization characteristics of quantization aware training, (2) gradient sparsification, (3) new compile time optimizations with Tiny Training Engine. Results are shown on several tinyML applications.

**Questions:**

Please refer to the last section.

**Limitations:**

Mostly yes. Other things like point 2 above (see weaknesses) can help further.

**Strengths And Weaknesses:**

The paper has several strengths:

1. Training on-device is much more difficult than inference on-device. Hence, doing simple transfer learning kind of tasks on tiny devices is interesting.

2. The paper makes significant contributions towards optimization of quantized networks as well as training support compilers for tiny devices.

3. The results demonstrate significant memory savings compared to standard deep learning frameworks (Tensorflow/Pytorch).

Despite the strengths, the paper does have several significant weaknesses:

1. One of the main motivations that authors use to disregard several existing training time/adaptation techniques (e.g, the ones that rely on updating only the normalization layers (reference [25] in paper)) is that the normalization layers are not present once the model is deployed on tiny devices. This happens because deployment tools like TFLITE, etc., fuse the batchnorm layers into convolution weights. However, this fusing process is extremely cheap, and the compiler needs to do it only once for inference. Can we not keep the norm layers for the on-device training purposes and fuse them only when running inference? Would training only the norm layers result in significant savings compared to the proposed approach? This is a major class of upcoming cheap training time/adaptation techniques and solid evidence must be presented to show that the proposed approach indeed outperforms this new class of models. The baselines considered for comparison (“update last k layers”, “update last k biases”, “update only the classifier”) are not strong. If “update only norm layers” kind of techniques are effective, it may also reduce the motivation behind the proposed quantization scaling method.

2. The improvements to the optimization problem for quantization aware training seem general. However, the authors have not provided any evidence that this would result in general improvements in quantization-aware training (and not just for tinyML applications). Specifically, would the proposed scaling method reduce the gap between FP32 and INT8 models for non-transfer learning scenarios? This kind of experiment can clarify whether the proposed technique is limited to transfer learning only or does it apply to general quantization-aware training? If it only works for transfer learning and not in general, why? Some discussion around specific conditions under which the proposed method works would strengthen the paper.

3. There are many existing gradient sparsification techniques. Authors have not presented any comparison against existing gradient sparsification techniques. For instance, this paper and the many references it cites are relevant: https://arxiv.org/pdf/1712.01887.pdf. Some newer papers probably exist in this space. The above paper is from 2017. I would recommend authors to do a thorough comparison against SotA gradient sparsification techniques.

4. Will the code for tiny training engine be open sourced?

Overall, I like the idea in this paper and the fact that this is the first paper that trains models on-device for realistic applications under 256KB. This is a significant contribution. However, I would like to make sure that the paper is not missing significant baselines and comparisons. I can increase the rating if the above weaknesses are adequately addressed.


---Update after rebuttal---
I have read other reviews and author response. The authors have provided good clarification and insightful new experiments during the rebuttal. They have satisfactorily answered my questions. As a result I have increased the rating to "6: Weak Accept". If the paper is accepted, I would like the authors to just show a simple comparison between standard quantization-aware-training (QAT, e.g., where BNs are still present and are fused after QAT is complete) and QAS on real quantized graph (e.g., the one without BN). While I can see that on a real quantized graph, QAS has benefits for non-transfer learning scenarios (based on new results), it would be useful to see the gap between regular QAT and QAS on real quantized graphs. Indeed, this can help motivate future works that may improve upon QAS to close the gap between QAT and quantization algorithms that directly work on real quantized graphs.

---

> ### Author Response · Authors · 2022-08-02
> **Author Response**
>
> We thank reviewer JJZV for the informative discussion. The comments and questions are very relevant.
>
> **Q1: Compare with only fine-tuning the Batch Normalization layers.**
>
> BN-only update is only **parameter**-efficient, but not **memory**-efficient. Training only the norm layers will not lead to further memory savings compared to our proposed approach.
>
> For the BN-only update, the number of trainable parameters is small. However, we need to save the intermediate activations to calculate the gradients of the BN parameters. Consider the scale $s$ and shift $b$ in BN: $y=sx + b$, to update $s$, we need to save the intermediate activation $x$ ($ds=x * dy$). The intermediate activation is much larger than the parameters of BNs and becomes the major memory bottleneck for transfer learning [12]. While our sparse update scheme allows us to skip a large part of the intermediate activation storage, leading to better memory saving.
>
> To further address your concerns, we experimented with updating the BN layers of MobileNetV2-w0.35. The results are added to supplementary Section J. As shown in the trade-off curve (can be viewed [here](https://anonymous.4open.science/r/on-device-training-0FE3/assets/figures/compare_bn_only.png)), BN-only update achieves a best average accuracy (on 8 datasets) of **66.4%** at **414kB** memory (OOM), while our sparse update scheme achieves **67.4%** at **49kB**, which is **1% higher on accuracy** with **8.4x smaller memory usage**. Our sparse update consistently outperforms BN-only update in both memory efficiency and accuracy.
>
> **Q2: Is QAS general for non-transfer learning scenarios?**
>
> Thanks for the suggestion. QAS can be applied as long as we are updating a **real** quantized graph, where parameters are quantized to integers (also no BNs). However, quantizing a randomly initialized model is not feasible since we cannot reliably measure the weight and activation range before training [33]. To tackle the issue, we first perform warm-up training for a small number of iterations (<5 epochs) to get a reasonable weight and activation range; and then perform model quantization to get the real quantized graph. We then train on the downstream tasks w/ and w/o QAS. Experiments show that **QAS is effective for the non-transfer learning scenario**: training w/ QAS leads to an average accuracy of **65.4%** on 8 datasets, while training w/o QAS can only achieve **48.3%**, as shown below. QAS effectively improves the training results by 17% for non-transfer learning scenarios. Please find the detailed results in supplementary Section K.
>
>  |Method  | Average Accuracy |  Cars     |CF10      |CF100      |CUB    |Flowers      |Foods | Pets    |VWW |
> |---------------------|:------------:|:------------:|:------------:|:------------:|:------------:|:------------:|:------------:|:------------:|:------------:|
> | SGD-M   | 48.3 | 15.8   | 63.5   | 41.0   |30.3    |69.8     | 34.1     |52.5      | 79.2      |
> | SGD-M+QAS  | **65.4** | 52.0    |85.5     |58.7      |47.1   |78.2    |53.1     |63.3      |85.6   |
>
> **Q3: Compare with gradient sparsification techniques like DGC.**
>
> Thanks for the suggestion. We would like to clarify that our sparse update is fundamentally different from gradient sparsification from both the motivation and implementation perspectives:
>
> - Gradient sparsification is designed for large-scale distributed training to save **communication costs** between servers, while our sparse update focuses on tiny-scale on-device learning to handle the limited **memory and computation**.
> - Gradient sparsification methods like DGC [a] perform gradient pruning **after** all the gradient is computed; thus it **cannot save** memory or computation (potentially memory overhead due to sorting and accumulation).
> - Our sparse update applies sparsification at the compile time **before** gradient computation. It can skip the less important gradient computation and intermediate activation storage, leading to real memory saving and speed up (e.g., 17.9x memory saving and 3x speed up for MobileNetV2-w0.35).
>
> Due to space limitations, we will reference and compare with the paper in the final version.
>
> [a] Deep Gradient Compression: Reducing the Communication Bandwidth for Distributed Training
>
> **Q4: Will the Tiny Training Engine code be open sourced?**
>
> Yes, we will open source the implementation upon publication. We have cleaned and uploaded the code to reproduce the video demo of transfer learning to the VWW dataset on a microcontroller (see supplementary Section A): [https://anonymous.4open.science/r/on-device-training-0FE3](https://anonymous.4open.science/r/on-device-training-0FE3). We hope the code will help the community reproduce our work and inspire later studies.
>
> We hope our response has resolved all of your concerns. Please let us know what other experiments or clarifications we can offer to convince you to increase the rating.

---

> > ### Comment · Reviewer_JJZV · 2022-08-07
> > **Thanks for the response**
> >
> > I have read other reviews and author response. The authors have provided good clarification and insightful new experiments during the rebuttal. They have satisfactorily answered my questions. As a result I have increased the rating to "6: Weak Accept". If the paper is accepted, I would like the authors to just show a simple comparison between standard quantization-aware-training (QAT, e.g., where BNs are still present and are fused after QAT is complete) and QAS on real quantized graph (e.g., the one without BN). While I can see that on a real quantized graph, QAS has benefits for non-transfer learning scenarios (based on new results), it would be useful to see the gap between regular QAT and QAS on real quantized graphs. Indeed, this can help motivate future works that may improve upon QAS to close the gap between QAT and quantization algorithms that directly work on real quantized graphs.

---

> ### Author Response · Authors · 2022-08-06
> **Author Response**
>
> Dear reviewer JJZV,
>
> Thanks again for your insightful suggestions and comments. As the deadline for discussion is approaching, we are glad to provide any additional clarifications that you may need.
>
> In our previous response, we added the following results:
>
> 1. The codebase to reproduce our on-device training demo;
> 2. Comparison between BN-only update and our method;
> 3. Study of QAS under non-transfer learning scenarios;
> 4. Comparison with gradient sparsification.
>
> We hope the provided new experiments and additional explanations have convinced you of the merits of our work. Please do not hesitate to contact us if there are other clarifications or experiments we can offer.
>
> Thank you for your time again!
>
> Best,
>
> Authors

---

### Official Review · Reviewer_Dz6f · 2022-07-12

**Rating:** 6
**Confidence:** 4
**Soundness:** 3 good
**Presentation:** 3 good
**Contribution:** 3 good

**Summary:**

This paper presents a quantization-aware-scaling mechanism to perform on-device training with extremely limited memory bandwidth settings. The system leverages sparse update throug Tiny Training Engine by pruning backward computation graph. The experiments on Visual wakeword demonstrates the efficacy of the proposed system.


**Questions:**

Q: The evaluation is done on a rather limited dataset, and the author should do more evaluation on a broader range of dataset to make it more convincing?

Ablation does not seem to have ablated QAS VS w/o gradient scale?




**Ethics Review Area:**

["I don’t know"]

**Limitations:**

I don’t see any statement of opensourcing the implementation, without such promise, I felt this work would be extremely difficult to reproduce. This is a glaring issue with the quantization community, so I would really see the code opensourced some day.

**Strengths And Weaknesses:**

This is a very well-written paper with extensive descriptions and many practical “tricks” to enable on-device training. Namely, QAS, sparse tensor update, DCE, in-place update…The demo shows a functioning working system on the real microcontroller. The work presented in the paper contains significant effort.

The individual tricks themself are not new, e.g. for the idea of QAS, I think the authors should cite LSQ<Esser et al. 2019>. On the whole, to enable all the parts work together systematically, this is non-trival effort.

The authors seem to have neglected <HFP8, Sun et al. 2020> and their follow up 4-bit ultra low precision work for comparison.

---

> ### Author Response · Authors · 2022-08-02
> **Author Response**
>
> **Q1: The comparison between QAS and LSQ (Esser et al., 2019)**
>
> We have cited the LSQ paper in the revised version. Our paper has a **fundamentally different** setting compared to LSQ; QAS also **consistently outperforms** the scaling method in LSQ in our setting (73.5% vs. 39.8% accuracy).
>
> Setting-wise:
> 1. LSQ is doing QAT on a **fake** quantization graph: the full precision weights need to be stored and updated during training, which costs 4x larger memory than int8 weights and does not lead to training memory saving. In fact, this is the setting of most existing papers aiming to train an accurate quantized model for efficient **inference** where our goal is efficient **training**.
> 2. In our setting, we are updating a **real** quantized graph, where all the weights and biases are integers from the very beginning and stay as integers after updates. We are not targeting the quantization process itself (like LSQ); instead, we are _given_ a quantized graph and want to update it for transfer learning.
> Please also kindly refer to supplementary Section D for the comparison between real quantized graphs and fake quantized graphs.
>
> We perform experiments to compare QAS with the scaling method in LSQ:
>
> 1. QAS consistently outperforms LSQ scaling. QAS can achieve an average accuracy of 73.5% on 8 datasets, while LSQ scaling only achieves 39.8% despite grid tuning on hyper-parameters.
> 2. QAS is based on the mathematical derivation of the quantization process, while LSQ scaling is based on a heuristic to keep the same weight/gradient scale across parameters. We think such a heuristic does not apply to the **real** quantized graph with very different tensor ranges (e.g., int32 has a $10^7\times$ larger range than int8) and leads to unstable training. This is a unique challenge when we update the real quantized graph and has not been observed in previous work.
>
> **Q2: Compare with HFP8 and the follow-up FP4 work.**
>
> We have cited the two papers in the revised version. The two papers target a fundamentally different setting to ours. The two papers use hybrid and customized FP8 or FP4 formats, which are only supported in specialized hardware. It cannot run on general-purpose hardware like MCUs, Arm CPUs, Intel CPUs, etc. While our work focuses on int8 quantization, which can efficiently run on general-purpose hardware and has wider applications like tinyML.
>
> We kindly remind the reviewer that QAS is just one part of our methods. Sparse update and training engine optimization bring **much larger memory saving** (17.9x, 1.9x in Figure 1), which should not be neglected. Limiting the scope to quantization does not provide a holistic view of our work.
>
> **Q3: Regarding using “tricks” to enable on-device learning.**
>
> Our proposed methods are fundamental principles instead of simple tricks. Our framework is the **first** solution to actually enable tiny on-device training of CNNs under a 256KB memory budget. We believe our design principles will shed light on later studies.
>
> 1. We are targeting a new setting of updating the **real** quantized graph instead of using QAT to train a fake quantized graph for inference. Existing optimization methods will lead to inferior accuracy while QAS successfully addresses the difficulties.
> 2. Sparse layer/tensor update is a novel memory-efficient update method. It consistently outperforms existing work like BN-only update [25], bias-only update [12], fine-tuning last k layers, etc. in terms of accuracy-memory trade-off.
> 3. Efficient training systems on edge are rarely explored. Our Tiny Training Engine provides fundamental designs like compile-time differentiation, backward graph pruning, op reordering, etc. to improve memory and computation efficiency.
>
> **Q4: Experiments on a broader range of datasets.**
>
> We have thoroughly verified the effectiveness of our proposed method on 8 datasets: Cars, CIFAR-10, CIFAR-100, CUB, Flowers, Food, Pets, and VWW, where our method achieves consistent improvement. These datasets are widely used to benchmark transfer learning in previous work (e.g., [13, 39]). So we believe the results we provided are comprehensive. Note that for the experimental results in the paper, we report the **average** accuracy on the 8 datasets to reflect the overall performance.
>
> **Q5: Ablation study on QAS vs. w/o gradient scaling.**
>
> We have already provided the ablation study in Table 1, comparing vanilla SGD-M (i.e., w/o gradient scaling) and SGD-M+QAS. QAS consistently improves the average learning accuracy on 8 datasets (73.5% w/ scaling v.s. 64.9% w/o scaling).
>
> **Q6: Statement of open sourcing.**
>
> Our method is fully reproducible. We will open source the code upon publication. We have uploaded the [code to reproduce the video demo](https://anonymous.4open.science/r/on-device-training-0FE3) of transfer learning to the VWW dataset on a microcontroller (see supplementary Section A). We hope the code will help the community reproduce our work and inspire more later studies.

---

> ### Author Response · Authors · 2022-08-09
> **Author Response**
>
> Dear Reviewer Dz6f,
>
> Thanks again for your insightful suggestions and comments. We have not heard from you and the rebuttal window is going to close. We believe our rebuttal should help clarify the concerns about related work, ablation studies (more datasets, QAS vs. w/o gradient scaling), and the statement of open sourcing. Please let us know if you have more questions.
>
> Bests,
>
> Authors

---

> > ### Comment · Reviewer_Dz6f · 2022-08-09
> > **Thank you for following up!**
> >
> > Dear Authors,
> > Thank you for your response!
> > Based on your replies and your promise of opensourcing the code. I am raising my score to 6.
> > Good luck!

---

### Author Response · Authors · 2022-08-02
**General Response**

We sincerely appreciate all reviewers’ time and efforts in reviewing our paper and for the constructive feedback. In addition to the response to specific reviewers, here we would like to thank reviewers for their acknowledgment of our work and highlight the new results added during the rebuttal:

We are glad that the reviewers appreciate and recognize our contributions:
- First work to train vision models on microcontrollers, together with a live demo [Dz6f, JJZV]
- Support the training of real quantized graphs with a novel scaling method [Dz6f, JJZV]
- Complete system-level support and significant memory saving over existing frameworks [HCuR, JJZV]
- Sufficient and detailed experiments to support the proposed method [JJZV, HCuR, xaNn, Dz6f]
- Well-written and easy to understand [Dz6f, HCuR]

In this rebuttal, we have added more supporting results following the reviewers’ suggestions.

- Open source code to reproduce the on-device training demo [JJZV, Dz6f]
- Comparison between BN-only update and our sparse update [xaNn]
- Train-from-scratch results with QAS [xaNn, JJZV]
- Speedup evaluation of sparse update on Qualcomm mobile CPU, Jetson Nano, and Raspberry Pi [xaNn]

---

### Author Response · Authors · 2022-08-06
**General Response**

Dear AC and reviewers:

Thanks again for all of your constructive suggestions, which have helped us improve the quality and clarity of the paper.

Since the discussion phase started four days ago, we have not heard any post-rebuttal response yet. Please don’t hesitate to let us know if there are any additional clarifications or experiments that we can offer, as we would love to convince you of the merits of the paper. We appreciate your suggestions. Thanks!

---

### Meta-Review · Area_Chair_UwwY · 2022-08-29

**Recommendation:** Accept
**Confidence:** Certain

**Metareview:**

In this work the authors propose a framework for training CV models on tiny IoT devices with very limited memory. The reviewers agreed that the paper is well written and represents a valuable contribution to the area of efficient / on-device ML. Questions raised by reviewers were sufficiently addressed in the response.

**Award:**

No

---

### Decision · Program_Chairs · 2022-09-14

Accept